# Serum and Glucocorticoid-Inducible Kinase 1 (SGK1) in NSCLC Therapy

**DOI:** 10.3390/ph13110413

**Published:** 2020-11-22

**Authors:** Ilaria Guerriero, Gianni Monaco, Vincenzo Coppola, Arturo Orlacchio

**Affiliations:** 1Biogem Institute for Genetic Research Gaetano Salvatore, Ariano Irpino, 83031 Avellino, Italy; ilaria.guerriero@biogem.it (I.G.); mongianni1@gmail.com (G.M.); 2Department of Cancer Biology and Genetics, College of Medicine, The Ohio State University, Columbus, OH 43210, USA; 3Arthur G. James Comprehensive Cancer Center, The Ohio State University, Columbus, OH 43210, USA

**Keywords:** NSCLC, SGK1, chemotherapy, immunotherapy

## Abstract

Non-small cell lung cancer (NSCLC) remains the most prevalent and one of the deadliest cancers worldwide. Despite recent success, there is still an urgent need for new therapeutic strategies. It is also becoming increasingly evident that combinatorial approaches are more effective than single modality treatments. This review proposes that the serum and glucocorticoid-inducible kinase 1 (SGK1) may represent an attractive target for therapy of NSCLC. Although ubiquitously expressed, SGK1 deletion in mice causes only mild defects of ion physiology. The frequent overexpression of SGK1 in tumors is likely stress-induced and provides a therapeutic window to spare normal tissues. SGK1 appears to promote oncogenic signaling aimed at preserving the survival and fitness of cancer cells. Most importantly, recent investigations have revealed the ability of SGK1 to skew immune-cell differentiation toward pro-tumorigenic phenotypes. Future studies are needed to fully evaluate the potential of SGK1 as a therapeutic target in combinatorial treatments of NSCLC. However, based on what is currently known, SGK1 inactivation can result in anti-oncogenic effects both on tumor cells and on the immune microenvironment. A first generation of small molecules to inactivate SGK1 has already been already produced.

## 1. Introduction

Lung cancer was the most commonly diagnosed cancer worldwide in 2018, accounting for 11.6% of new total cancer cases (14.5% in males and 8.4% in females) and causing about 1,700,000 deaths (18.4% of all cancer-related deaths) [1]. Based on its histopathological features, lung cancer has been categorized into small-cell lung carcinoma (SCLC), which represents 15% of all lung cancer cases, and non-SCLC (NSCLC), which accounts for the remaining 85%. NSCLC is further classified into three subgroups: adenocarcinoma (LUAD), squamous cell carcinoma (LUSC), and large-cell carcinoma (LACC) [2]. Finally, these different types of NSCLC subgroups have been classified according to the WHO guidelines, which were revised in 2015 [3]. In detail, LUAD can be divided in three prognostic groups: lepidic pattern (good prognosis), acinar and papillary pattern (intermediate prognosis), and micropapillary and solid pattern (worse prognosis) [4]. LUSC, instead, can be categorized in three histological groups: keratinizing, non-keratinizing, and basaloid form, according to the correlation between keratinization and clinical outcome [5].

LACC has neither clear features of LUAD and LUSC nor expression of neuroendocrine markers. Generally, LACC tumors are considerably undifferentiated and included in this group because of exclusion from the previous ones [6].

All NSCLCs are generally characterized by cellular subpopulations with distinctive molecular and histological features that require a “personalized medicine” type of treatment. Despite significant improvement due to the introduction, in the clinics, of novel treatments with small molecules inhibiting tyrosine kinases and immunotherapy, NSCLC remains a deadly disease, especially when invasion and metastases develop [7]. The unmet medical need for curative therapeutic alternatives can be overcome only by a deeper understanding of the unknown mechanisms underlying tumor progression, including the relationship between cancer cells and the tumor microenvironment.

In recent decades, a number of genetic alterations and oncogenic driver mutations have been identified in NSCLC, thus emphasizing the heterogeneous nature of this disease [8]. In this regard, phosphoinositide 3-kinase (PI3K) is the most deregulated pathway in cancer, with a broad pathological impact [9,10]. Although protein kinase B (AKT) is classically considered the main effector of the PI3K signaling cascade, recent growing evidence is suggesting that other proteins impinging upon this pathway or intersecting with it are playing a critical role during neoplastic transformation independently of AKT [11]. These players are also involved in the establishment of resistance to PI3K/AKT inhibitors [12,13] and several of them have become targets of therapy. Targeted drugs against epidermal growth factor receptor (EGFR), anaplastic lymphoma kinase (ALK), ROS proto-oncogene 1 (ROS1), Ki-ras2 Kirsten rat sarcoma viral oncogene homolog (KRAS), B-Raf Proto-Oncogene (BRAF), human epidermal growth factor receptor 2 (HER2), rearranged during transfection (RET), and MET, are now successfully used in clinics [14,15,16,17,18]. Several of these compounds have distinctly improved the outcome of NSCLC treatment [19].

There are a variety of genetic and epigenetic alterations that can negatively impact the efficacy of a treatment regimen. They can affect the primary target of the drug or other proteins, which can activate pathways parallel or downstream in respect to the original target, thus overcoming its direct inhibition [20,21,22,23,24,25].

Serum and glucocorticoid-inducible kinase 1 (SGK1) is a member of the AGC kinase family of serine/threonine kinases. Some of the most notable members of this family are AKT, 3-phosphoinositide-dependent kinase-1 (PDK1), Ribosomal S6 kinase p70 (S6K), Protein Kinase C (PKC), and ribosomal s6 kinase p90 (RSK). Studies that aimed to elucidate the biological functions and the specific targets of phosphorylation of each AGC kinase have been hampered by the high degree of sequence and structural homology observed in this family. Indeed, SGK1 and AKT do share a large homologous sequence and several targets [26,27]; however, differently from AKT, SGK1 does not possess a pleckstrin homology (PH) domain and, therefore, it cannot directly interact with phosphatidylinositol 3, 4, 5 tris-phosphate [28].

SGK1 is activated by a two-step process. First, a phosphorylation on Ser422 performed by the mammalian target of rapamycin complex 2 (mTORC2) induces the kinase to assume an open conformation. Full activation is achieved through a second phosphorylation event operated by PDK1 on SGK1 Thr256. Specifically, PDK1 PIF pocket is responsible to recognize SGK1 after it has been primed by mTORC2 [29,30]. This is another important difference with AKT, which does not require PDK1 PIF pocket to be activated [31].

Among the physiological processes regulated by SGK1 are ion transport, embryo implantation and pregnancy, T-cell activation, macrophage motility and function, insulin sensitivity, and apoptosis [32,33,34,35,36,37,38]. It is, then, no surprise that SGK1 has been implicated in a variety of diseases [39,40,41], including in different types of human cancer [42,43,44].

This review focuses on the potential targetability of SGK1 in NSCLC specifically. It is beyond our scope to systematically list of all the potential or verified targets of SGK1 phosphorylation. On the other hand, we aim to highlight pro-tumorigenic signaling networks and biological processes that the activation of SGK1 mediates directly or indirectly in NSCLC tumors. Of particular interest are the effects that SGK1 activation has in cells of the immune system to promote their differentiation toward pro-tumorigenic subpopulations.

## 2. The SGK Protein Family

The SGK family consists of three different SGK paralogs: SGK1, SGK2, and SGK3 [45]. They are encoded by three distinct genes, each one localized on a different chromosome [46].

SGK1, SGK2, and SGK3 proteins share a highly homologous structural organization, containing an N-terminal variable domain, a catalytic domain, and a hydrophobic domain at the *C*-terminal (Figure 1).

The three paralogs show the lowest degree of homology at the N-terminal domain [47,48].

SGK3 is the only family member with a pleckstrin homology (PH) domain at the N-terminus, which is important for its membrane localization [33,49]. SGK2 has a shorter N-terminus and is expressed only in liver, pancreas, kidney, and, at lower levels, in brain [47]. SGK1 and SGK3 are ubiquitously expressed [48].

Each SGK paralog has more than one splicing isoform. However, the current knowledge about the functional relevance of each one of those variants is still limited.

Four isoforms of SGK1 have been described. They all differ at the N-terminal domain due to alternative use of different translation initiation sites [50]. As a functional consequence, SGK1.2 and SGK1.3 are rapidly degraded through the 26S proteasome because of a hydrophobic motif at the N-terminal domain, which is missing in the other two variants [50]. This motif is responsible for the localization of SGK1 at the endoplasmic reticulum (ER) [51], where it can be ubiquitinated by the stress-associated chaperone-dependent E3 ligase (CHIP) [52]. The SGK1.1 isoform is not only less susceptible to ubiquitination by lacking the N-terminal ER-homing motif but also highly expressed in brain and pancreas. Furthermore, like SGK2 and SGK3, SGK1.1 is not sensitive to glucocorticoid stimulation [53]. Another important characteristic of this isoform is the cellular localization. In fact, SGK1.1 can bind phosphatidylinositol 4,5-bisphosphate though a cluster of positively charged and hydrophobic residues [54,55]. When levels of phosphatidylinositol 4,5-bisphosphate drop, SGK1.1 moves to the cytosol and accumulates in the nucleus thanks to the same aminoacidic cluster which is also a nuclear localization signal [54,55].

Both SGK2 and SGK3 have two variants, SGK2α and SGK2β, which differ at the N-terminus, while the SGK3.1 and SGK3.2 isoforms are the result of an alternative exon exclusion event.

Like with SGK1.4, the functional relevance of SGK2 and SGK3 isoforms is still unknown. However, the existence of three SGK genes and multiple variants is a confounding factor in the identification of targets and functions specific to SGK1.

## 3. Mouse Models to Elucidate SGK Protein Functions

Mouse models have provided critical evidence in establishing what the effects of SGK protein depletion are in vivo.

Interestingly, despite its ubiquitous expression, the genetic inactivation of SGK1 in mouse causes only mild phenotypes related to ion physiology. In fact, SGK1 knockout mice are viable but show a decreased ability to retain salt when exposed to a salt-deficient diet [56], an altered renal response to K^+^ load [57], and a decreased renal ability to excrete Ca^2+^ [58]. Salt depletion may also explain why those mice show increased plasmatic aldosterone concentrations [57]. Further, SGK1^−/−^ mice exposed to either a high-fructose or high-fat diet show resistance to hypertension [59,60] and are less responsive to the effect of mineralocorticoids on both salt appetite [61] and intestinal glucose uptake [62]. The latter is also reduced in brain cells, adipocytes, and skeletal muscle [63]. Finally, SGK1^−/−^ mice show a mild defect of platelet migration [64].

The deletion of SGK3 in mouse does not cause a strong phenotype either. SGK3^−/−^ mice may have a transient delay both in hair growth and body weight increase [65]. Only a more detailed analysis revealed that SGK3 deletion results in reduced intestinal glucose transport and a very mild locomotion defect [66].

Interestingly, SGK1/SGK3 double-knockout mice show a compound phenotype that combines those observed in the two single mutants [67].

SGK2^−/−^ mice do not show any phenotype. However, the analysis of mice lacking both SGK1 and SGK2 has revealed that SGK2 bears a certain degree of functional redundancy with SGK1 in contributing to water and electrolyte homeostasis during salt deprivation [68].

In summary, the engineering of different SGK-deficient animals has shown that there are biological and cell-type functions specific to SGK1 that neither AKT nor SGK2 and SGK3 are able to compensate for [69]. On the other hand, these in vivo observations have also suggested that inhibition of SGK1 could be safely achieved.

## 4. SGK1 Expression

SGK1 expression is proven to be tightly regulated both transcriptionally and post-transcriptionally [32,33,70,71].

SGK1 is ubiquitously expressed and it is the only family member that was reported being regulated by glucocorticoids [47]. Growth factors, cytokines, and insulin are among the diverse types of stimuli that can affect SGK1 expression [42,72,73] (Table 1).

SGK1 is functionally connected to cellular stress and, unlike the other two paralogs, can be phosphorylated on S78 by BIM-K1 [74] and p38-MAPKinase [75]. Different types of cellular stress have effects on SGK1 expression, including DNA damage and heart ischemia [76,77] (Table 1).

Interestingly, transcript levels do not always correlate with amount of protein. In fact, some tissues show high expression of SGK1 when probed for mRNA but a low protein amount. Other than artifacts in the methods used for detection of SGK1 mRNA and protein, this is also, in part, attributed to the short half-life of the kinase active form (~30 min) [91].

Based on two independent studies that assessed a possible interaction by surface plasmon resonance (SPR), it has been suggested that the ubiquitin ligase Nedd4-2 targets SGK1 for degradation by the 26S proteasome [92,93]. This hypothesis is supported by the observation in HEK-293T and COS-7 cells that Nedd4-2 overexpression leads to decreased SGK1 levels while Nedd4-2 silencing results in the increase in the SGK1 half-life [91,94,95]. On the other hand, SGK1 ubiquitination and subsequent degradation are also promoted by a Rictor/Cullin E3 ligase complex [96,97]. Interestingly, AGC kinases such as SGK1, AKT, and S6K can phosphorylate Rictor on T1135. This phosphorylation is reported to impair the interaction between Rictor and Cullin-1, resulting in the inhibition of SGK1 ubiquitination and degradation and establishing a positive feedback signaling loop [96,97].

The glucocorticoid-induced leucine zipper protein-1 (GILZ1) [98] is reported to stabilize SGK1 by opposing its localization to the ER and, therefore, limiting its interaction with ER-based E3 ubiquitin ligases, such as CHIP [52] and Synoviolin 1 (HRD1) [91].

## 5. SGK1 Is a Predicted Target of microRNAs Relevant in NSCLC

MicroRNAs (miRNAs) are an integral part of the complex mechanisms tightly regulating SGK1 expression at the post-transcriptional level. They are deregulated in cancer and generally exert oncogenic effects through the silencing of their target genes. NSCLC patients frequently show altered expression of several miRNAs that have been associated with the progression of disease and resistance to therapy [99,100,101,102,103]. Several of those miRNAs that are known to be deregulated in lung cancer are also predicted to target SGK1 [104].

For a group of miRNAs that are downregulated in NSCLC, the literature is quite consistent.

miR-497 is down-regulated in several human malignancies, suggesting a tumor-suppressive role [105]. In NSCLC cell lines and patient specimens, miR-497 has been found downregulated and has been reported to inhibit cell proliferation in vitro and tumor growth in vivo by targeting the hepatoma-derived growth factor (HDGF) [106]. Moreover, miR-497 is able to function as a tumor-suppressor miRNA in NSCLC by targeting the vascular endothelial growth factor A (VEGF-A) [107], cyclin E1 [108] and the Yes-associated protein 1 (YAP1) [109,110]. Due to the complicated interaction with its many downstream targets, miR-497 is currently being studied more as a diagnostic marker rather than a therapeutic target [105]. However, it is interesting to note that this miRNA has also been reported to target AKT2, impairing tumor growth and chemoresistance in lung cancer cell lines and xenograft models [111]. If its ability to target SGK1 will be experimentally validated, this miRNA could be a tool to inhibit both kinases at the same time.

miR-15 was identified for the first time in 2002, together with miR-16, as a potential oncomiR in the pathogenesis of chronic lymphocytic leukemia (CLL), being frequently deleted and/or downregulated in this malignancy [112]. Recently, Yang et al. performed a meta-analysis to assess the prognostic value of miR-15a in human cancers [113], confirming that this miRNA is frequently downregulated, in particular in NSCLC [114]. Moreover, miR-15a has been found to regulate epithelial-to-mesenchymal transition (EMT) by silencing Twist1 [115], a key regulator of this process, together with Slug, which has a relevant role in lung carcinogenesis [116]. Interestingly, miR-15a has been proposed as a prognostic biomarker in NSCLC since is able to reduce the expression of several oncogenic proteins [117]. Furthermore, decreased levels of miR-15a have been associated to cisplatin resistance in NSCLC [118]. The induced overexpression of this miRNA is able to turn back on apoptosis and autophagy in cancer cells treated with cisplatin, possibly by targeting Bcl-2 [119]. SGK1 has been shown to play a role in this process, since its inhibition, by either GSK650394 or SGK1 shRNA, may induce G2/M arrest, apoptosis, and autophagy through the mTOR-FoxO3a pathway in other human cancers [120,121].

miR-181a/b. In recent years, different groups have proposed the use of miR-181a/b in new therapeutic approaches to treat NSCLC [122]. Indeed, miR-181b has been found increased in stage I NSCLC while its expression is significantly reduced at later stages [123,124,125,126,127,128]. miR-181b overexpression inhibits tumor cell proliferation, migration, invasion, and metastatic colonization. Therefore, the progressive down-regulation of this miRNA is a powerful way to increase cancer aggressiveness. Interestingly, miR-181a plays a role in the function of tumor-associated macrophages, showing higher expression in the M2 immunosuppressive phenotype compared to M1 [129]. Since the immune contribution is crucial in the tumor microenvironment to sustain the progression of the disease, and taking into account that the M2 macrophages localize in the hypoxic regions of lung tumors, miR-181a could have a critical role through specific targets, such as SGK1, which is reported to be involved in the regulation of macrophage polarization [130,131] and is also activated by hypoxic stimuli [130,132,133].

miR-125. miR-125a and 125b share a seed sequence and possibly the same targets. miR-125a is another one of those miRNAs generally down-regulated in many human cancers, including NSCLC [134]. It has been shown to function as a tumor suppressor by targeting signal transducer and activator of transcription 3 (STAT3) [135] and several other downstream effectors of the KRAS and nuclear factor-kappa B (NF-kB) pathways, often together with miR-23b [136]. Both miR-125a and miR-125b are involved inflammatory processes. Their delivery by bone marrow mesenchymal stem cell-derived exosomes ameliorates the symptoms of colitis in mice through the silencing of STAT3. At the same time, they have been shown to inhibit Th17 cell differentiation while favoring regulatory T cell (Treg) expansion [137]. Interestingly, SGK1, which is predicted to be targeted by miR-125, is similarly involved in Th17 and Treg cells’ regulation (see below).

Finally, a group of miRNAs that are predicted to target and downregulate SGK1 [104], such as miR-96 [138,139,140], miR-183 [141,142,143], miR-130b [144,145,146], miR-182 [147,148,149,150], miR-301b [151,152], miR-17 [153,154], and miR-9 [155,156,157,158], are overexpressed in NSCLC. However, although they have not been validated yet, the fact that SGK1 is frequently upregulated suggests that mechanisms allowing escape from the miRNA-mediated suppressive effect might be in place. On the other hand, these miRNAs target other molecules that directly or indirectly cooperate with SGK1 signaling. Therefore, miRNAs can add an additional layer of complexity to the regulation of SGK1 expression. Future investigations about their interconnection in the context of NSCLC could lead to a better understanding of the pathophysiology of the disease and new theragnostic tools.

## 6. SGK1 Is a Prognostic Factor in NSCLC

SGK1 is frequently upregulated in NSCLC. Both its transcript and protein expression have been related to prognosis [159,160,161,162]. In 2012, Abbruzzese et al. found that higher SGK1 mRNA expression in NSCLC patients correlated with worse prognosis, supporting an oncogenic role of SGK1 [160]. The authors also analyzed the expression of each SGK1 splice variant separately. However, such an analysis produced data that were not as statistically significant as when considering the sum of the four variants. High SGK1 mRNA expression was detected in mainly LUSC, but it needs to be noted that the study was not able to find a correlation with prognosis when the protein levels of SGK1 were measured. There may be several possible explanations for this discrepancy. For example, the different half-lives of the four splicing variants and the possible effect of specific miRNAs could result in translational rather than transcriptional regulation. On the other hand, it is also possible that the measurement of protein expression by immunohistochemistry is far less precise than quantitative PCR and, therefore, the cohort of samples (n = 66) used for the study may have been underpowered.

A recent study by Pan et al. assessed the expression of SGK1 by immunohistochemical detection using a larger cohort of 150 cases consisting uniquely of human lung adenocarcinoma. In contrast to Abbruzzese et al., the authors found a significant correlation between high SGK1 protein levels and increased nodal invasion and advanced disease (stage III/IV). On the other hand, low levels of SGK1 correlated with the absence of nodal metastasis. No significant correlation of SGK1 levels with age, sex, smoking history, histopathologic grade, or T stage was found [161].

In 2018, Tang et al. also correlated SGK1 protein expression with several clinical parameters of NSCLC patients [162]. The authors reported a significant association between high SGK1 protein expression and differentiation or histological type, but not with lymph node metastasis and advanced TNM staging. More importantly, their analysis showed that high expression of SGK1 was a significant negative prognostic factor for 5-year survival.

Since SGK1 expression is related to cellular stress, it is plausible and, in many cases, demonstrated, that high expression can be found in high-grade/advanced tumors, which are frequently characterized by enhanced metabolism and excessive oxidative stress due to the “Warburg effect” [130,132,133,163]. In addition, external stress stimuli can also produce DNA damage with the consequent activation of P53 that, in turn, can activate the transcription of SGK1 [162] (see below).

## 7. Potential Targeting of SGK1 in Combination with Chemotherapy to Treat NSCLC

Despite the recent success in the management of NSCLC, there is still an urgent need for innovative treatment strategies and new potential targets. Chemotherapy is still used in most patients. Usually, platinum-based agents (cisplatin or carboplatin) are used in combination with other drugs, such as gemcitabine, vinorelbine, paclitaxel, or taxotere [164]. Finally, corticosteroids are often used as a co-treatment in advanced lung cancer [165].

Resistance to treatment is a major common limitation of the current therapeutic modalities [166]. The display of intrinsic resistance is evident at the initial treatment due to innate tumor features usually of a genetic or epigenetic nature. Most commonly, tumors acquire resistance due to adaptive advantageous adaptation or clone selection [166].

Along the same lines, it is widely accepted that single modalities of treatment are rarely curative, while combinatorial strategies are much more efficacious. Hence, due to the vast signaling network that SGK1 operates, its inactivation can result in synergistic effects together with targeted- and chemotherapy. Theoretically, the elevated levels that SGK1 shows in NSCLC patients provide a window of treatment that may spare normal tissues. This is in addition to the fact that total body inactivation in mice only causes mild phenotypes.

P53 is key in the response to chemotherapy, which generally induces DNA damage. SGK1 has recently been linked with resistance to DNA damage-induced apoptosis in ERCC excision repair 1 (ERCC1)-defective lung cancer cells, suggesting an indirect involvement with specific types of DNA damage [167]. In this regard, SGK1 is known to phosphorylate mouse double minute 2 homolog (MDM2) and, therefore, affect the levels and the regulatory activity on proliferation and cell survival of P53 [168]. In turn, P53 is known to positively regulate the transcription of SGK1, establishing a negative feedback loop between the two molecules [76,169,170]. Moreover, P53 degradation can also be promoted through Nedd4-2, which is an SGK1 target [171]. This observation also suggests that antiapoptotic effects exerted by SGK1 may be delivered through the regulation of P53, at least in some cases.

One pivotal cellular player protecting from apoptosis, including when induced by DNA damage, is the nuclear factor-kappa B (NF-κB). Constitutive NF-κB activation has been reported in different kinds of tumors [172,173]; more importantly, high levels of NF-κB activation have been found in tumor samples from lung cancer patients, both for SCLC and NSCLC, and it is associated with poor prognosis [173,174]. Indeed, NF-κB inhibition has been shown to negatively affect lung cancer cell survival and proliferation [175,176,177]. In cancer cells, NF-κB activation is also known to be induced by chemotherapeutics and radiation, thus contributing to resistance [178,179]. On the other side, it has been shown that NF-κB inhibition can increase the efficacy of both therapeutic approaches in vitro and in vivo [180,181]. SGK1 can activate NF-κB signaling though two different mechanisms. On the one hand, SGK1 phosphorylates the IκB kinase β at Ser177 and Ser181, thus promoting its degradation and, consequently, NF-κB nuclear translocation. On the other hand, SGK1 directly phosphorylates p300 at Ser1834, thus enhancing NF-κB acetylation and activity [182]. Therefore, targeting of SGK1 could constitute an alternative to direct NF-κB inhibition, with the potential advantage of limiting the side effects observed with that approach [178,183,184,185,186,187].

SGK1 seems to be at the crossroad between PI3K and Wingless-related integration site (WNT) signaling. SGK1 has been reported to interact with Parvin Alpha (PARVA), leading to the activation of Integrin Linked Kinase (ILK) with the consequent phosphorylation of AKT and the glycogen synthase kinase 3β (GSK3β) (Ser9) [188]. Interestingly, a recent article has shown that SGK1 depletion leads to impaired growth and migration of NSCLC cells [159]. More importantly, the authors found that SGK1 knockdown leads to a lower level of β-catenin, confirming that SGK1 can inactivate GSK-3β and cause accumulation of β-catenin.

This result not only opens new avenues of investigation but also shows how SGK1 promotes one of the major oncogenic pathways, such as the Wnt/β-catenin signaling. It has been established that WNT signaling plays a role in drug resistance in NSCLC [189,190,191]. Indeed, it has been reported that activation of the Wnt/β-catenin signaling pathway by inhibiting cytoplasmic GSK-3β induces increased resistance to cisplatin in A549 cells [192]. Using the same cell model, Zhang et al. reported that knockdown by small interfering RNA (siRNA) of β-catenin increases sensitivity to cisplatin by decreasing both mRNA and protein levels of the antiapoptotic factor B-cell lymphoma-extra large (Bcl-xl) [193]. However, Wnt/β-catenin signaling is now recognized as a major determinant of the “immune exclusion” phenotype of lung tumors by decreasing the infiltration of immune system killer cells and reducing the efficacy of immune checkpoint inhibitor (ICI)-based therapy [194,195].

In a perspective of cancer treatment, an interesting area of investigation in relation to the potential therapeutic effects provided by targeting of SGK1 is offered by the control exerted by this kinase on a variety of channels involved in ion and nutrient absorption [196]. The mild phenotypes observed in SGK KO mouse models are in line with these channel-related functions. Most importantly, several of those channels have independently been proposed, themselves, as candidate targets of therapy in lung cancer.

The chemical and genetic inactivation of Kv1.3 resulted in a significant reduction in proliferation in A549 cells both in vivo and in vitro [197]. Overexpression in the same cell line of the acid-sensing ion channel 1 (ASIC1) promotes proliferation and migration induced by extracellular acidosis. On the other hand, chemical and genetic inhibition of ASIC1 reduced proliferation and migration of A549 cells [198].

Other SGK1-controlled targets are the Ca^2+^ release-activated Ca^2+^ channels (ICRAC). They are composed of the ORAI1, -2, and -3 subunits, which form the actual pore, and the regulatory elements stromal interaction molecule (STIM) -1 and -2. SGK1 both negatively affects ORAI1 degradation by repressing the ubiquitin ligase Nedd4-2 and increases the transcription of ORAI1 and STIM1 through NF-kB [196,199]. Recent reports show that ORAI1 is associated with poor prognosis in NSCLC and its knockdown negatively affects PI3K/AKT/ERK signaling [200].

From a therapeutic point of view, STIM1 knockdown has been shown to inhibit proliferation both in vivo and in vitro in NSCLC cell lines [201]; however, there are contrasting reports on its effect on cisplatin-induced apoptosis [202,203].

Finally, another SGK1 target, the epithelial sodium channel (ENaC), and specifically its alpha subunit, is a direct transcriptional target of the oncogene achaete-scute homolog 1 (ASCL1) in lung neuroendocrine tumors [204].

## 8. Potential Targeting of SGK1 in Combination with Immune-Therapy to Treat NSCLC

Tumor development is the result of a tug-of-war between cancer cells and the surveillance exerted by the immune system. Reactivation and boosting of the immune system anticancer response using checkpoint inhibitors (ICIs) against the programmed cell death protein 1 (PD1), the programmed death-ligand 1 (PD-L1), and the cytotoxic T-lymphocyte-associated protein 4 (CTLA4) have now set new standards for treatment of NSCLC, especially in cases without an identified driver mutation [205]. Patient response to those drugs can be dramatic, with spectacular results at times. However, about half of the cases do not respond to ICIs to begin with and most responses are only temporary. Due to the fact that SGK1 has been associated with drug resistance in several types of cancer [206], targeting of SGK1 in combination with ICIs can provide a potential synergy because of both tumor cell-intrinsic and -extrinsic effects.

Corticosteroids are used in advanced NSLC to mitigate the inflammation associated with the tumors [165]. However, these drugs can exert a global tumor-promoting effect because of tumor cell-intrinsic and tumor cell-extrinsic actions.

Specific to NSCLC, exposure to dexamethasone of adenocarcinoma cell lines induces inhibition of apoptosis by upregulation of SGK1 [207]. Since SGK1 is downstream of the PI3K–AKT signaling axis, its glucocorticoid-induced upregulation could explain the appearance of resistance to PI3K–AKT by “opportunistic compensation” [13].

Mechanistically, it is not yet understood how SGK1 exerts its oncogenic role. However, it has been proposed that SGK1 might, at least in part, promote cancer proliferation, differentiation, migration, invasion, and resistance to alkylating chemotherapy through the activation of its best characterized downstream target, N-myc downstream regulated gene 1 (NDRG1) [208,209]. In NSCLC, the chromosomal region including NDRG1 is consistently amplified and the gene over-expressed [210] SGK1 can promote stem-like properties of lung cancer cells by preventing c-Myc degradation through ubiquitination operated by the S-phase kinase-associated protein 2 (Skp2). NDRG1 inactivates cyclin-dependent kinase 2 (CDK2), leading to reduced Skp2 phosphorylation [208]. Moreover, the levels of expression of NDRG1 in NSCLC patients are associated with poor prognosis and advanced T-stage [211,212].

SGK1 is also responsible of the inhibitory phosphorylation of the forkhead transcription factor FKHRL1 (also known as FoxO3a), thus inducing cell survival and cell cycle progression [169]. In NSCLC patients, FoxO3a expression has been observed to be significantly decreased in tumor tissues compared [213,214] with adjacent normal tissues. Evidence suggests that FoxO3a inhibits the expression of EGFR pathway substrate 8 (EPS8), preventing cancer progression [213]. Furthermore, FoxO3a is frequently deleted in early-stage lung adenocarcinoma and reported to stimulate apoptosis in response to DNA-damaging agents in adenocarcinoma cell lines [215].

All considered, the targeting of SGK1 could potentially eliminate tumor cell-intrinsic pro-tumorigenic effects by ablating its corticosteroid-induced upregulation and signaling.

Targeting of SGK1 has the potential to affect the anti-tumor immune response significantly and positively through other mechanisms too.

As mentioned above, SGK1 knockdown in tumor cells can lead to lower levels of β-catenin. WNT/beta catenin signaling is now accepted to be one major factor in determining an “immune-exclusion” phenotype where lymphocytes are kept out of tumor masses [194,195]. Therefore, SGK1 targeting can be beneficial in combating immune exclusion.

Targeting of SGK1 is also predicted to affect immune system cells directly. In addition to the aforementioned potential role in promoting macrophage differentiation towards the M2 pro-tumorigenic phenotype, GSK1 may also promote T-cell differentiation toward tumor-promoting subpopulations. Indeed, for an effective immune response, the balance between specific types of T-cells, such as that of T-helper1 and T-helper2 (Th1/Th2) or that of Th17 and regulatory T cells (Treg), has been shown to be critical. The Th1/Th2 ratio is being extensively studied not only in NSCLC microenvironment but in other cancers as well [216,217,218,219,220,221].

Th1 cells produce interleukin (IL)-2, interferon-gamma (IFN-γ), and tumor necrosis factor-alpha (TNF-α), and seem to be the main effectors of host immune response against intracellular pathogens. Th2 cells, on the other hand, produce IL-4, IL-5, IL-6, IL-10, and IL-13, and are involved in allergic responses and protection against certain parasitic infections [222,223].

By phosphorylating GSK3β and Nedd4-2, SGK1 can skew the differentiation of Th1/Th2 cells.

In 2014, Heikamp et al. demonstrated that SGK1 is essential for the differentiation of CD4^+^ T cells into Th2 helper cells [35]. Specifically, SGK1 phosphorylation of Nedd4-2 inhibits JunB ubiquitination, therefore stabilizing it. This, in turn, activates a gene expression pattern involving IL-4 and GATA-3, which favors the differentiation in Th2 cells. At the same time, by inhibiting GSK3β and stabilizing β-catenin, SGK1 induces the expression of T cell factor 1 (TCF1), which represses the transcription of genes associated with the Th1 phenotype, such as INF-γ. The same authors also reported that mice lacking SGK1 specifically in CD4^+^ helper precursor cells showed decreased IL-4 and increased levels of IFN-γ and were able to mount an effective immune response to B16 melanoma cells in comparison to wild-type controls. This is in line with the role of IFN-γ in affecting the activity of natural killer cells, macrophage activation, and antigen presentation and promoting cancer cell elimination by engaging tumor-specific CD8⁺ cytotoxic T lymphocytes [219,224].

It is worth noting that GILZ, which is known to stabilize SGK1, when overexpressed in CD4^+^ cells, can have opposite effects when compared to SGK1 deletion. In fact, GILZ transgenic mice showed an increase in IL-4 and a decrease in IFN-γ levels [225].

Targeting of SGK1 can also alter the balance between Th17 and Treg cells, which has also been found to be important, specifically in NSCLC [226,227,228]. SGK1 favors the Th17 lineage by increasing IL-23 signaling though inhibitory phosphorylation of FoxO1, ultimately relieving its repressing activity on IL-23 receptor expression [229,230]. FoxO1 can also affect Treg by controlling the expression of Foxp3 [231]. Th17 and Treg cells are generally thought to play opposing roles in regulating immunity [232]; however, while their balance is critical for maintaining homeostasis, their functions have been shown to be context-dependent, allowing them to play dual roles in cancer, and specifically in NSCLC [233,234,235].

Ultimately, more studies are necessary to both elucidate the role of SGK1 as a regulator of immune cell differentiation and to investigate its potential as target to improve cancer immunotherapy. Immunocompetent mouse models should be used for this aim.

## 9. SGK1 Inhibitors

The search for molecules inactivating SGK1 has already started. To date, only a handful of ATP-mimetic compounds have been demonstrated to be selective inhibitors of SGK1 (Figure 2).

While these compounds did help to elucidate some of the SGK1-specific downstream targets, their use in clinical settings has been limited because of low specificity, modest bioavailability, and reduced cell permeability.

The most used SGK-1 inhibitor, GSK650394, was described for the first time in 2008 by Sherk et al. [236]. Chemically, it is a heterocyclic derivative of indazole with relatively poor cell permeability [237]. In testing its specificity, the authors reported that the observed IC_50_ was similar for SGK1 and SGK2 (62 and 103 nM, respectively). Moreover, this compound has been found to have off-targets in subsequent studies. In particular, it has been shown to be only less than 10-fold more selective for SGK1 than for Aurora and c-Jun N-terminal kinase and only 30-fold for IGF1R, ROCK, JAK1 and JAK3, AKT1/2/3, DYRK1A, and PDK1 [238].

In 2011, a benzohydrazide derivative, later labeled as EMD638683, was described by Ackermann et al. [239] as an SGK1 inhibitor with an in vitro IC_50_ of 3 μM. The authors also evaluated it for in vivo use. However, the effective dose needed was rather high (600 mg/Kg). EMD638683 seemed to be slightly more selective than GSK650394 in terms of off-targeting. However, both compounds needed to be used at 10 μM or higher concentrations to observe a significant decrease in NDRG1 phosphorylation. Additionally, EMD638683 inhibits MSK1 and PRK2 with an efficiency similar than that of SGK1, SGK2, or SGK3 [239].

More recently, Ortuso et al. reported the identification of a novel SGK1 inhibitor now named SI113. Based on the Pyrazolo-Pyrimidine Scaffold, SI113 showed remarkable selectivity for SGK1 when tested in comparison with AKT1. Although preclinical reports suggested low toxicity, this inhibitor has not been commercialized yet [206,240].

Halland et al., in 2015, described N-[4-(1H-Pyrazolo [3,4-b]pyrazin-6-yl)-phenyl]-sulfonamides as SGK1 inhibitors and, in particular, the compound “14 g”, which was renamed one year later as “SGK1-inh” by Castel et al. Although this compound has a reported IC_50_ of 4.8 nM for SGK1 inhibition, its use is significantly limited by poor cell permeability. Consequently, high doses are required to achieve a significant effect on NDRG1 (>10 μM) [12,241,242].

Lastly, QGY-5-114-A was developed as an analog of GSK650394 and has recently been reported by Liang et al. to have a reasonably low IC_50_ and to show promising in vivo results in xenograft models [243].

What is important for the identification of compounds capable of inhibiting SGK1 is the development of specific activity assays. Currently, the effect of SGK1 activity candidate inhibitors has been assessed with methods broadly used for other kinases, usually by measuring [γ-^32^P] ATP incorporation into a substrate peptide [12].

An interesting variation of the same principle was applied by Bezzerides et al., who developed an assay based on a phosphorylated peptide labeled with a green, fluorescent dye. SGK1 activity was, in this case, assessed by measuring the increase in the value of fluorescence polarization upon binding with a phospho-specific antibody [244].

As evidence of the relevance of SGK1 in cancer increases, more inhibitors are expected to be developed for testing in the clinics. The biggest obstacle to achieve this goal does not appear to be specificity of targeting, but better pharmacodynamics.

## 10. Conclusions

Although identified and initially characterized in 1993 [70], SGK1 is only recently being recognized as an essential mediator in a variety of processes, far more diverse than originally thought.

Most importantly, SGK1 is overexpressed in cancer and consistent evidence suggests that SGK1 can be a prognostic factor and a potential therapeutic target for the treatment of NSCLC. However, further studies are required to better assess its role in NSCLC pathogenesis and treatment response. While it is certainly true that efforts should be made towards the identification of new selective inhibitors of SGK1 suitable for in vivo studies, it is even more important to fully unravel the intricate network of functional interactions that this kinase operates in. To this end, the use of SGK1 conditional mouse models will be paramount to clarify direct and specific SGK1 targets. Their use will be essential to define cell autonomous vs. non-cell autonomous SGK1 functions connected with NSCLC development and progression.

## Figures and Tables

**Figure 1 pharmaceuticals-13-00413-f001:**
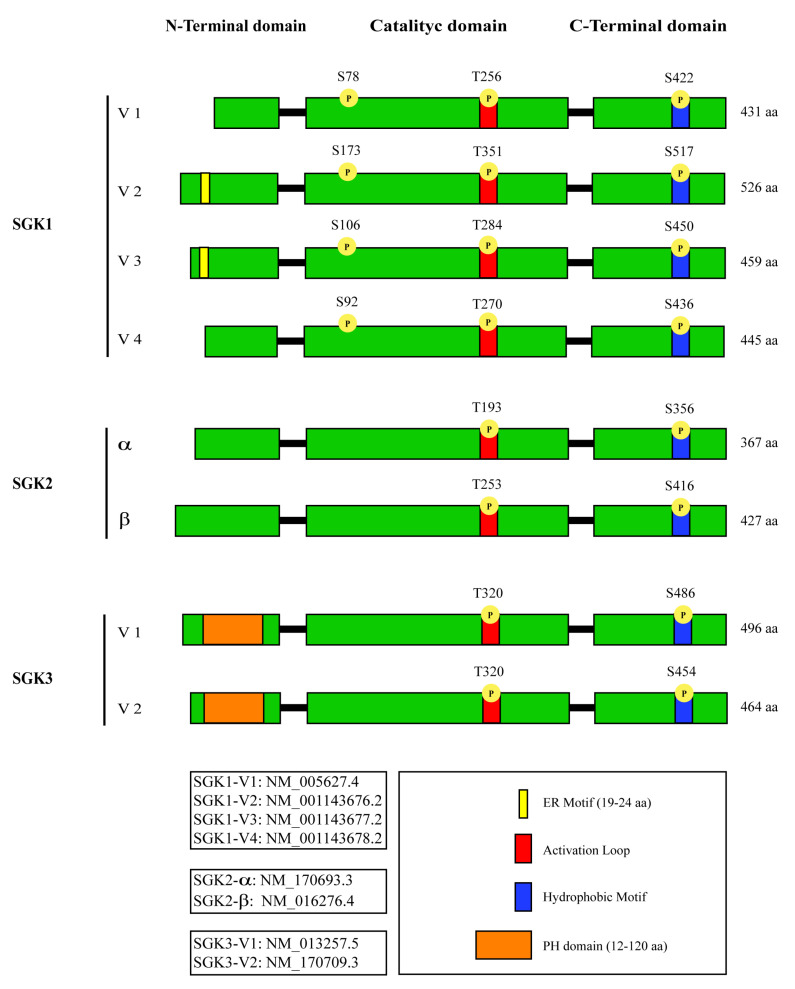
Schematic representation of known serum and glucocorticoid-inducible kinases SGK1, SGK2, and SGK3 splicing variants.

**Figure 2 pharmaceuticals-13-00413-f002:**
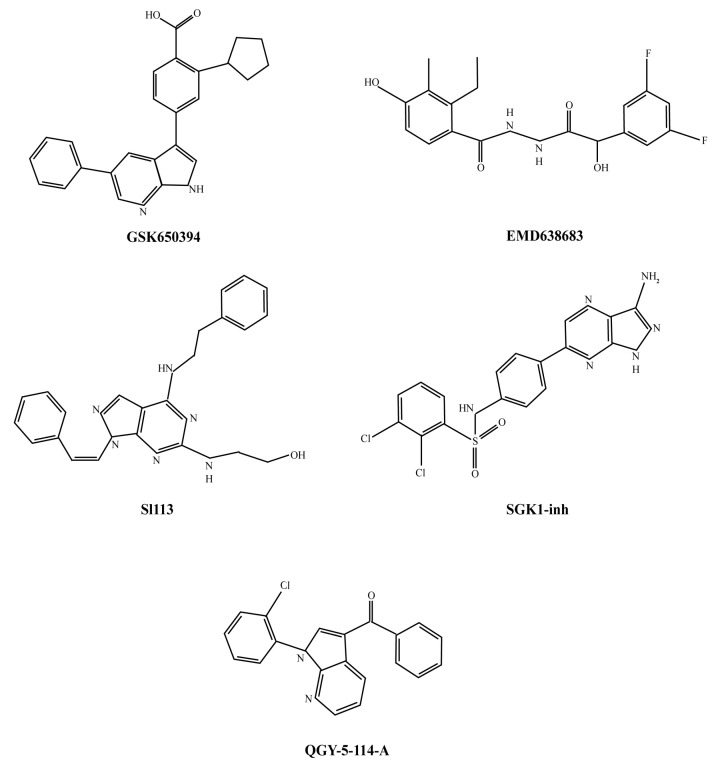
Chemical structure of small molecules capable of inhibiting SGK1.

**Table 1 pharmaceuticals-13-00413-t001:** Known regulators of SGK1 expression and activity. Abbreviation: Interleukin-2 (IL-2), Internleukin-6, tumor necrosis factor alpha (TNF-α), Fibroblast growth factor (FGF), Transforming growth factor beta (TGF-β), Platelet-derived growth factor (PDGF), Follicle-stimulating hormone (FSH), Luteinizing hormone (LH), Insulin-like growth factor 1 (IGF1).

Type of Stimuli	Regulators of SGK1	References
Cytokines	IL-2	[78]
	IL-6	[75]
	Colony-Stimulating Factor 2	[79]
	TNF-α	[79]
Growth Factors	Glucocorticoids	[70]
	Mineralocorticoids	[80]
	FGF	[72]
	Serum	[70]
	TGF-β	[81]
	PDGF	[72]
	FSH	[82]
	LH	[83]
	Insulin	[84]
	IGF1	[26]
Cellular stress	Osmotic stress	[85]
	Heat-shock	[86]
	UV	[86]
	Ischemic injury	[87]
	Hepatitis	[88]
	Sorbitol	[89]
	Hydrogen peroxide	[47]
	Neuronal injury	[87]
	High Glucose	[90]

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
