# Peer review of "Serum and Glucocorticoid-Inducible Kinase 1 (SGK1) in NSCLC Therapy"

_pharmaceuticals, 2020, doi:10.3390/ph13110413_

Round 1

Reviewer 1 Report

Serum and Glucocorticoid- Inducible Kinase 1 (SGK1) in NSCLC 2 Therapy

Comments 

  1. "SGK1 deletion in mice does not cause major phenotypes" please change to give more clear information. I think in a review it is more important to present data as is. 
  2. prognostic factor: are there any report for this SGK1 as I am curious to see elaborated in this review. 
  3. Please also comment on the availability of activity methods (HTS, SPR or biochemical or fluorescence-based assays for SGK1. 

overall a very interesting article and written very well and can be published as is. 

Author Response

We would like to thank the reviewer for its comments and for the time dedicated to our manuscript.

Please find below our point-by-point response.

1."SGK1 deletion in mice does not cause major phenotypes" please change to give more clear information. I think in a review it is more important to present data as is. 

We have now revised the quote in the abstract as follows: “SGK1 deletion in mice only cause mild defects of ion physiology”. The details of the phenotype are then discussed in detail in paragraph 3.

2. prognostic factor: are there any report for this SGK1 as I am curious to see elaborated in this review. 

Best of our knowledge, the only relevant articles suggesting the potential of SGK1 as a prognostic factor in NSCLC are discussed in paragraph 6. We are not aware of any other report on the matter. Therefore, we are not sure whether the reviewer is referring to additional articles that we did not mention. If that is the case, we will be happy to further revise the manuscript shall the reviewer clarify this point.

3. Please also comment on the availability of activity methods (HTS, SPR or biochemical or         fluorescence-based assays for SGK1. 

We thank the reviewer for this comment. This is an important point and the info will be useful to the interested readership. However, we had not included it in our review because the methods currently used to assess SGK1 activity are not different from the ones used other kinases like AKT, for example.

We have now added a small paragraph (in red, lines 520 - 527) where we describe the most used. We have also added reports in which SPR was used to elucidate SGK1 interactions (in red, lines 198 - 200).

Reviewer 2 Report

Your review is very interesting and attractive. 

Further, clinical study are awaited to better define the role of SGK1. 

Good job!

Author Response

Thank you again for your kind comments and for the time you dedicated to our manuscript.